# Large-scale rollout of extension training in Bangladesh: Challenges and opportunities for gender-inclusive participation

John William Medendorp[1], N. Peter Reeves[2], Victor Giancarlo Sal y Rosas Celi[3], Md. Harun-ar-Rashid[4], Timothy J. Krupnik[5], Anne N. Lutomia[1], Barry Pittendrigh[1,6], Julia Bello-Bravo[7] *

1 The Urban Center, Department of Entomology, Purdue University, West Lafayette, Indiana, United States of America, 2 Sumaq Life LLC, East Lansing, Michigan, United States of America, 3 Sección de Matemáticas, Departamento de Ciencias, Pontificia Universidad Católica del Perú, San Miguel, Perú, 4 Agricultural Advisory Society (AAS), Dhaka, Bangladesh, 5 International Maize and Wheat Improvement Center (CIMMYT), Dhaka, Bangladesh, 6 Department of Entomology, Michigan State University, East Lansing, Michigan, United States of America, 7 Department of Agricultural Sciences Education and Communication, Purdue University, West Lafayette, Indiana, United States of America

* mbellobr@purdue.edu

**Data Availability Statement:** The data underlying the results presented in the study are available from https://purr.purdue.edu/publications/3983/1.

## Abstract

Despite the recognized importance of women's participation in agricultural extension services, research continues to show inequalities in women's participation. Emerging capacities for conducting large-scale extension training using information and communication technologies (ICTs) now afford opportunities for generating the rich datasets needed to analyze situational factors that affect women's participation. Data was recorded from 1,070 video-based agricultural extension training events (131,073 farmers) in four Administrative Divisions of Bangladesh (Rangpur, Dhaka, Khulna, and Rajshahi). The study analyzed the effect of gender of the trainer, time of the day, day of the week, month of the year, Bangladesh Administrative Division, and venue type on (1) the expected number of extension event attendees and (2) the odds of females attending the event conditioned on the total number of attendees. The study revealed strong gender specific training preferences. Several factors that increased total participation, decreased female attendance (e.g., male-led training event held after 3:30 pm in Rangpur). These findings highlight the dilemma faced by extension trainers seeking to maximize attendance at training events while avoiding exacerbating gender inequalities. The study concludes with a discussion of ways to mitigate gender exclusion in extension training by extending data collection processes, incorporating machine learning to understand gender preferences, and applying optimization theory to increase total participation while concurrently improving gender inclusivity.

## Introduction

While progress has been slow, inclusion of women at all levels of development efforts is increasingly recognized as critical for sustainability [1–3], especially in the agricultural sector [4–7], where significant percentages of the workforce are women [2, 8–12].

**Funding:** This publication was made possible in part by internal funds from Michigan State University and Purdue University [JBB & BRP]. The fall armyworm animation and scaled extension activities were supported by funds provided to CIMMYT by the Borlaug Higher Education for Agricultural Research and Development Program (BHEARD; USAID award # AID-BFS-G-11-00002) at Michigan State University [JWM] and the Bill and Melinda Gates Foundation support for the Cereal Systems Initiative for South Asia (CSISA; USAID award #BFS-IO-17-00005 and BMGF isINV-009787)[TJK]. USAID: https://www.usaid.gov BMGF: https://www.gatesfoundation.org Its contents are solely the responsibility of the authors and do not necessarily represent the official views of USAID, BHEARD, CSISA, or the Bill and Melinda Gates Foundation. No sponsors or funders played any role in the study design, data collection and analysis, decision to publish, or preparation of the manuscript.

**Competing interests:** The authors have declared that no competing interests exist.

Since the agricultural sector is a major (if not the major) contributor to GDP in many developing countries—representing from 10% to 55% of its total GDP and averaging 22.1% for "low-income countries" [13, 14]—investments in inclusive agricultural training for women yields a potentially greater return on developmental efforts [1, 15].

Women's participation, however, is typically affected by gender inequalities that limit the reach of both traditional and the more recent ICT-based approaches to agricultural extension [16]. While women's participation in agricultural production is significant in developing countries [17, 18], sociocultural and structural barriers make access to formal and non-formal education, as well as extension services, more difficult for women [19]. Despite the felt need for women's participation in agricultural extension services, as evidenced by initiatives focused specifically on women's agricultural inclusion, many agricultural extension activities nonetheless do not intentionally pursue gender inclusion [20, 21]. Moreover, at the organizational level, there are comparatively fewer women extension agents, and gender exclusion within those services often discourages participation by women [22].

In addition to these explicit gender inequalities within extension services, other implicit socio-structural factors, including but not limited to the effects of cultural expectations and roles for women, can also impact female participation in training, as we discuss in detail later in this paper [20, 23]. The relatively recent emergence of scalable ICTs—such as culturally and linguistically adapted, computer-animated videos [23, 24]—enables large-scale, ICT-based extension training that reaches more people, including those in hard to reach and underrepresented groups.

However, this opportunity also comes with challenges. Misdirected or poorly-fitted development interventions risk decreasing rather than increasing women's participation [25, 26]. As such, any mass-scaled ICT approach that is not properly implemented could conceivably exacerbate the very gender inequalities it seeks to mitigate. For instance, ICT-based educational extension services that focus on increasing agricultural training in the general population could successfully lead to better attended events [27–29], but if these larger events are dominated by male attendance, women may be reluctant to participate, especially in traditional male-dominated environments, such as those commonly associated with agriculture.

Our objective for this study was to analyze the effects of situational factors—time of day, gender of trainer, day of the week, month of the year, Administrative District, and venue type—on participation in extension events in general, and women's participation in particular. Our research questions therefore were: 1) what factors increased overall participation in ICT-based extension events and, 2) What were the odds of women attending any given event?

## Materials and methods

This study represents an observational study using existing data recorded at extension events conducted throughout Bangladesh. This study was deemed exempt by Michigan State University Biomedical and Health Institutional Review Board (IBR 00004626).

### Intervention

The data sets for this study were collected from October, 2018, to January, 2019, by the Agricultural Advisory Society (AAS) in partnership with the International Maize and Wheat Improvement Center (CIMMYT). CIMMYT extension training agents showed digital agricultural education materials (described below) at various venues across four of the eight Administrative Divisions in Bangladesh, namely, Rangpur, Dhaka, Khulna, and Rajshahi (Fig 1). Video screening venues included educational institutions, farmers houses, marketplaces, religious institutions (mosques and temples), and roadside venues such as shops, and tea stalls. In

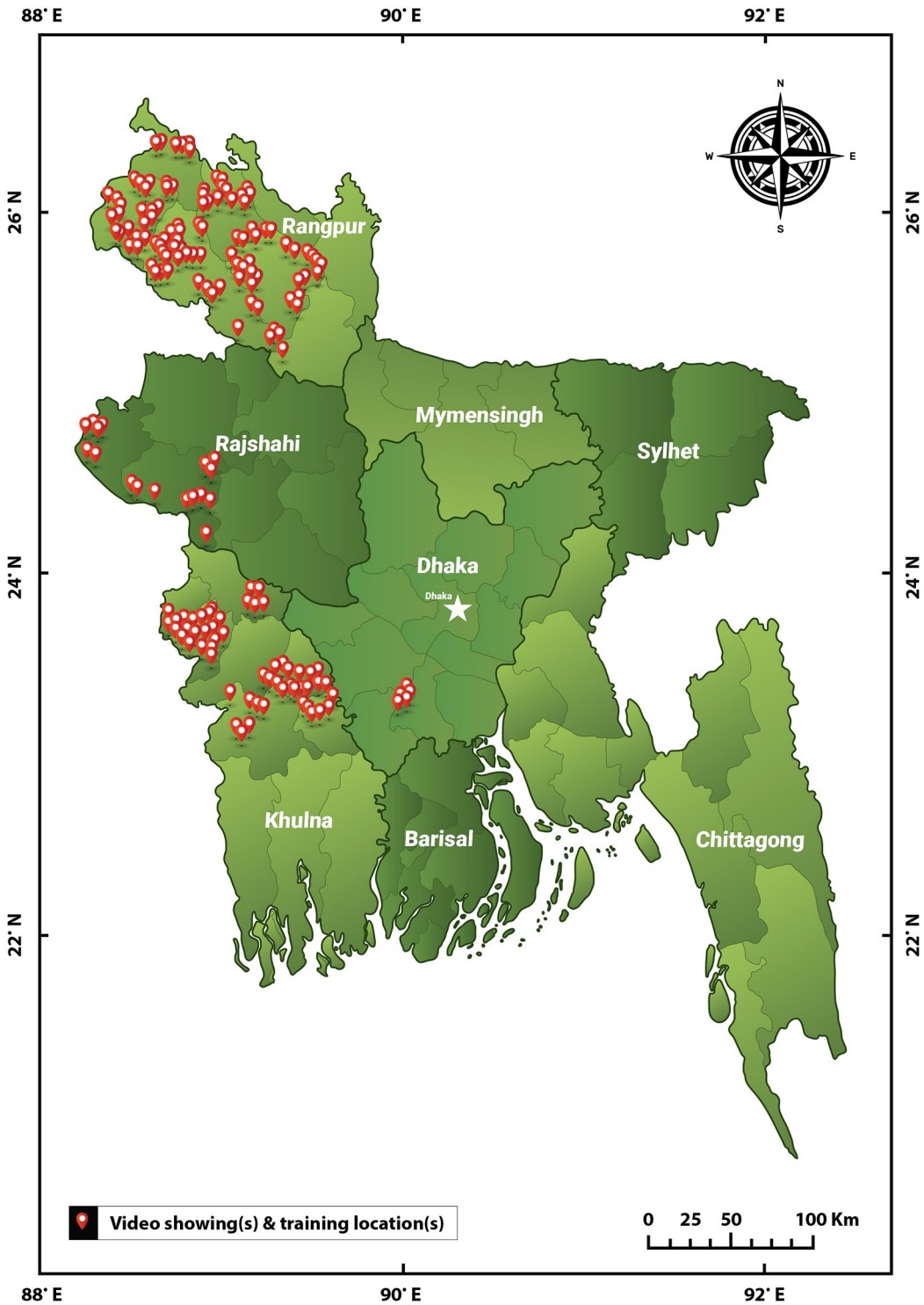

**Fig 1. (A) Agricultural extension services training sites in four of the eight national divisions of Bangladesh: Rangpur, Dhaka, Khulna, and Rajshahi and (B) the eight Administrative Divisions of Bangladesh (this file is made available under the Creative Commons CC0 1.0 Universal Public Domain Dedication at https://en.wikipedia.org/wiki/File: Bangladesh_divisions_english.svg).**

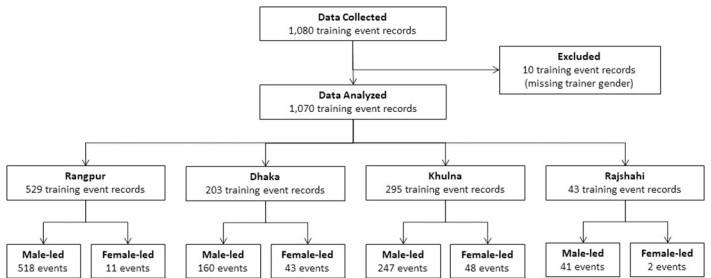

**Fig 2. Research design and implementation.**

addition, videos were shown at Union parishad campuses (i.e., governmental council buildings and grounds) and other public venues such as open spaces in front of hospitals, sport clubs, rail gates, bus stands, and playgrounds.

Experienced extension training agents showed one or more of the following educational agricultural videos at each of the training events:

1. an animated video for mitigating Fall Armyworm [30]; a major invasive maize pest that can cause up to 70% in yield losses [31, 32];

2. a non-animated video on how to plant healthy rice seedlings [33]; rice is a primary staple food that provides 48% of rural employment, about two-thirds of the overall supply of calories, and about half of the average person's total protein intake within the country, occupying around 74% of the total cropped area which accounts for 50% of Bangladesh's agricultural GDP and a sixth of its national income [5, 34];

3. a non-animated video advocating earlier planting dates to increase yields of wheat, which is Bangladesh's second most important food crop [35].

Locations for video projection were purposefully selected to maximize attendance in order to disseminate information on these topics in advance of the primary maize, rice, and wheat cropping seasons. At each event, three trained enumerators collected data on the total number of persons attending, the number of women attending, the gender and education level of the extension agents who administered the training, and time of the day, day of the week, month of the year, and the location and type of venue in which videos were shown. Only adults over the age of 18 were included in the data collection.

Among the 1,080 sessions conducted (132,358 attendees), 10 sessions had missing information on the gender of the trainer; therefore, only 1,070 sessions (131,073 attendees) were included in the analysis (see Fig 2 and Table 1).

## Data analysis

Statistics were calculated to obtain means, standard deviations, and ranges for continuous variables, and frequencies and percentages for categorical variables. As described, data was gathered based on registered participants in training sessions. Therefore, non-prior specification of data management was implemented.

To better understand situational factors that increase participation in the general population and among women, two statistical models were created. The two outcome variables corresponding to each model were (1) the total number of persons attending the event and (2) conditional on the number of persons attending the event, the odds of females attending the event. Odds of females attending was used instead of total number of females attending since

**Table 1. Descriptive statistics of the 1070 training events[b].**

| Characteristics | N (%) |
|---|---|
| Trainer that conducted sessions | |
| Gender (Female) | 104 (9.7) |
| Years of education [a] | 8 (2.5, 2–16) |
| Total number of trained persons [a] | 123 (103.7,15–600) |
| Total number of trained women [a] | 23 (25.8, 0–150) |
| Administrative Division | |
| Rangpur | 529 (49.4) |
| Dhaka | 203 (19.0) |
| Khulna | 295 (27.6) |
| Rajshahi | 43 (4.0) |
| Month | |
| October | 135 (12.6) |
| November | 299 (27.9) |
| December | 391 (36.5) |
| January | 245 (22.9) |
| Day of the week | |
| Sunday | 154 (14.4) |
| Monday | 143 (13.4) |
| Tuesday | 137 (12.8) |
| Wednesday | 147 (13.7) |
| Thursday | 155 (14.5) |
| Friday | 167 (15.6) |
| Saturday | 167 (15.6) |
| Time of day [a] | 3:00 PM (5.4, 7 AM–10:30 PM) |
| 7 AM–11 AM | 307 (28.7) |
| 11:01 AM–3:30 PM | 240 (22.4) |
| 3:31 PM– 6:00 PM | 356 (33.3) |
| 6:01 PM– 10:30 PM | 167 (15.6) |

[a] Mean (Standard deviation, Range)

[b] Ten sessions have the gender of the trainer missing.

female attendance is restricted based on total overall attendance, thus requiring a different statistical model for analysis. The explanatory variables considered were gender of the trainer, time of the day, day of the week, month of the year (ranging from October, 2018, to January, 2019), Administrative Division, and venue type. Since participants were not necessarily aware of the education levels of the trainer, this variable was considered as a confounder.

In order to assess situational factors associated with the expected number of persons attending any given event, a negative binomial regression model using a log link was fitted. Negative binomial, instead of a Poisson model, was fitted because the variance of the dependent variable was greater than the mean. On the other hand, in order to assess situational factors associated with the odds of females attending the event, given the total number of persons attending, a binomial regression model with a logit link was fitted.

Time of the day was categorized in four groups based on the observed quartiles, and years of education by the trainer was assessed using a cubic spline, since this variable was considered as a confounder. Venue types with less than 9 observations were grouped in an "others"

category. Nested models were compared using a likelihood ratio test. All tests were two-tailed and a 5% significance level was used. Statistical analyses were performed using R version 4.0.2.

## Results

### Descriptive data

The mean number of persons attending a given session was 123 (SD = 103.7), and the mean number of females attending a given session was 23 (SD = 25.8). The most common number of persons attending training was 41 (101 events representing 9.4% of total sessions), and for females it was zero (254 events representing 23.7% of total sessions) (Fig 3). In terms of trainers, only 104 (9.7%) were conducted by female extension agents. Most sessions were conducted in Rangpur (n = 529, 49.4%), followed by Khulna (n = 295, 27.6%) and Dhaka (n = 202, 19.0%), with the least being conducted in Rajshahi (n = 42, 4.0%). Events were distributed relatively evenly among the days of the week and occurred most frequently in the month of December.

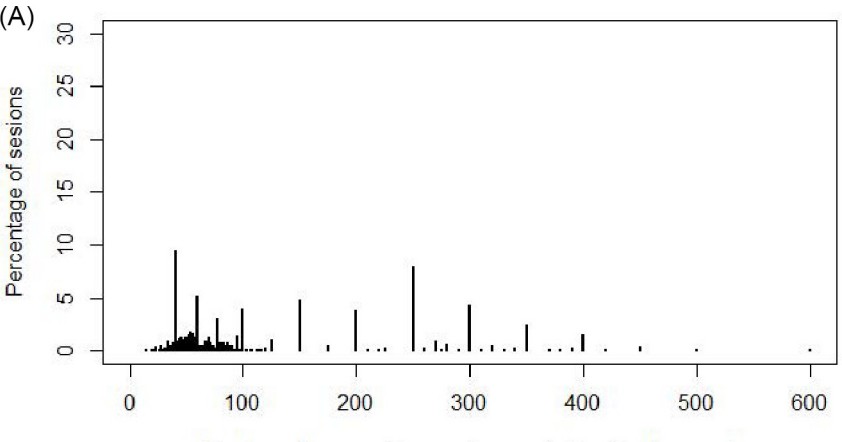

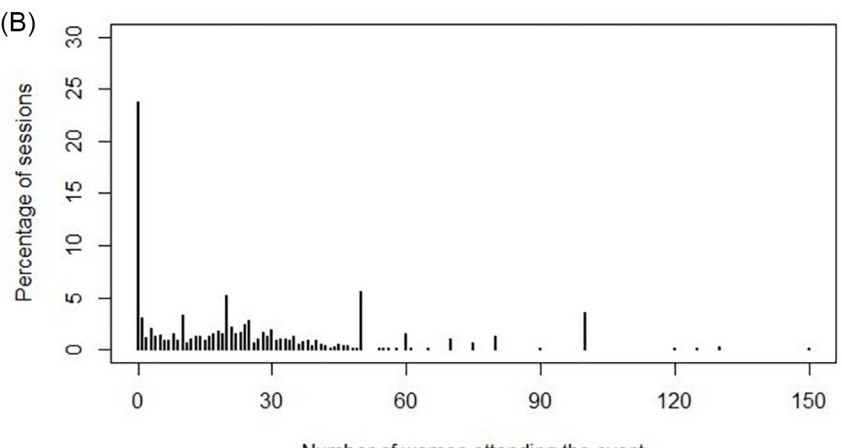

**Fig 3. (A) The distribution of number of persons (males and females) and (B) the distribution of number of females attending training sessions.**

## Main results

The first set of regression models considered time of the session in four groups (before 11:01 AM, between 11:01 AM–3:30 PM, between 3:31–6:00 PM, after 6:00 PM). However, a likelihood ratio test found no evidence (*p*-value = 0.604) that a four-group division was different than the one that considers only three groups, specifically: before 11:01 AM (morning), 11:01 AM–3:30 PM (mid-day), after 3:30 PM (late afternoon/evening). Therefore, the three-group division was used for analysis.

For the situational factor *time of the day*, the expected total number of persons attending a mid-day session was 33% lower (exp(β) = 0.67, 95% CI: 0.60–0.75) than a morning event. Furthermore, the expected total number of persons attending a late afternoon/evening event was 64% higher (exp(β) = 1.64, 95% CI: 1.47–1.84) than the expected total number of persons attending a morning event. In contrast, the odds of females attending a mid-day or late afternoon/evening event was 34% higher for a mid-day event (OR = 1.34, 95% CI: 1.28–1.41) and 27% lower for a late afternoon/evening event (OR = 0.73, 95% CI: 0.70–0.77) in comparison with a morning event.

For the situational factor *gender of the trainer*, the expected total number of persons attending a session was 16% higher (OR = 1.16, 95%CI: 1.01–1.33) if the trainer was male. Conversely, the odds of a female attending a session was 47% lower (OR = 0.53, 95%CI: 0.50–0.56) for male-led training events. Fig 4 shows general (Fig 4A) and female (Fig 4B) attendance within the context of the *gender of the trainer* and the *time of the day*.

For the situational factor *day of the week*, the expected number of persons attending a training session was not different amongst days; however, the odds of females attending a session was higher on most days in comparison with Sunday (the Bangladesh weekend is Friday and Saturday, with Friday being the day for religious services). More specifically, women were most likely to attend training sessions on Tuesdays (30% higher, OR = 1.30, 95% CI: 1.22 = 1.38) and Wednesdays (29% higher, OR = 1.29, 95% CI: 1.22–1.37), followed by Saturday (18% higher, OR = 1.18, 95% CI: 1.11–1.25), and Monday (17% higher, OR = 1.17, 95% CI: 1.10–1.24) compared to Sunday (See Table 2).

Regarding the *month in the year* in which the training events were conducted, the overall expected total number of persons attending a training session did not differ from October, 2018, to January, 2019, while the odds of females attending training during those months significantly increased over time with women being 82% (OR = 1.82, 95% CI 1.67–1.98) more likely to attend training in January than in October.

For the situational factor *Administrative Division*, taking Rangpur as a point of reference, the expected total number of persons attending a session was 36% lower in Dhaka (exp(β) = 0.64, 95% CI 0.55–0.75) and 23% lower in Khulna (exp(β) = 0.77, 95% CI 0.68–0.87) but similar in Rajshahi. On the other hand, the odds of females attending a session were 23% higher in Dhaka (OR = 1.23, 95% CI 1.14–1.33), 8% higher in Khulna (OR = 1.08, 95% CI 1.02–1.14), and 55% higher in Rajshahi (OR = 1.55, 95% CI 1.43–1.69) in comparison with Rangpur.

For the situational factor *venue type*, the expected total number of persons attending training was not significantly different between venues, except for trainings conducted at religious venues (35% lower, exp(β) = 0.65, 95%CI 0.49–0.87) and shops (26% lower, exp(β) = 0.74, 95% CI 0.57–0.95) compared to educational institutions. Importantly, however, the odds of females attending a session varied considerably between venues. Women were more likely to attend training in farmers' houses (388% more likely, OR = 4.88, 95% CI 4.52–5.28), religious places (355% more likely, OR = 4.55, 95% CI 4.05–5.11), in shops (177% more likely, OR = 2.77, 95% CI 2.44–3.03), and tea-stalls (144% more likely, OR = 2.44, 95% CI 2.23–2.67), while being less likely to attend training in a marketplace (69% less likely, OR = 0.31, 95% CI 0.29–0.34) and a

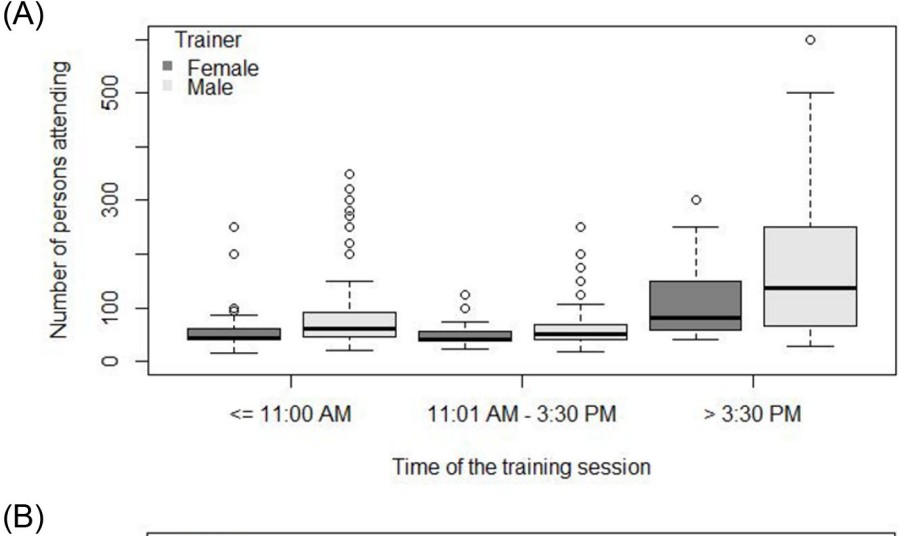

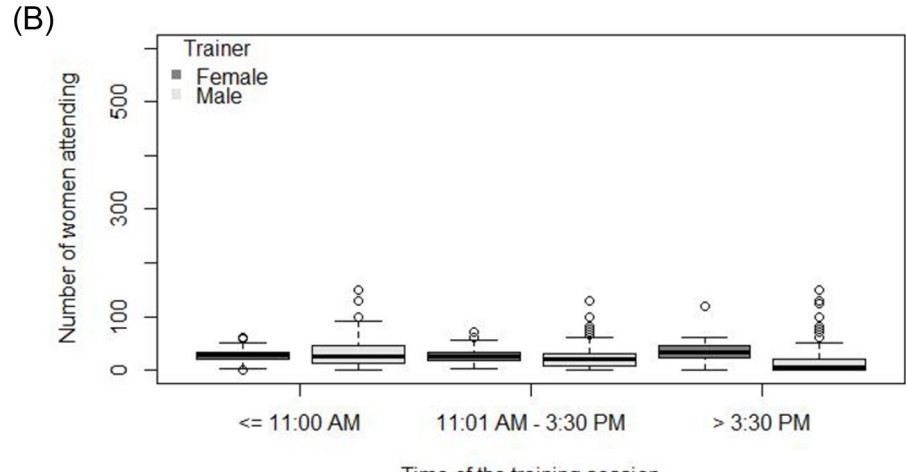

**Fig 4. (A) Total number of persons (males and females) and (B) total number of females attending training over the three time periods of the day, separated by male and female trainers.**

Union parishad campus (53% less likely, OR = 0.47, 95% CI 0.39–0.56), in comparison with educational institutions.

## Discussion

The results above provide critical insights into where and when Bangladeshi agricultural communities are more likely to access and participate in extension training services. In an era of chronic funding shortages for extension, such insights have the potential to provide more cost-effective strategies for when and where to stage extension trainings to improve the participation of all farmers while at the same time increasing the participation of women. Nonetheless, the noted differences in access and participation between the general population and women also create their own challenges for improving the reach of training, given that increasing overall attendance may negatively impact female attendance.

Such results are, however, perhaps not surprising in culturally conservative countries, including in Bangladesh. For example, many communities in rural areas of Bangladesh practice *purdah*, which is frequently translated as the practice of female exclusion from public

**Table 2. Estimates for the negative-binomial and binomial regression models to assess factors associated with the total number of persons and the odds of women attending the event.**

| | Negative binomial regression [a,b] | | | Binomial regression [a] | | |
|---|---|---|---|---|---|---|
| | exp(β) | 95%CI | *p*-value | OR | 95%CI | *p*-value |
| Time of day | | | | | | |
| ≤ 11:00 AM | Reference | | | Reference | | |
| 11:01 AM–3:30 PM | 0.67 | 0.60–0.75 | < **0.001** | 1.34 | 1.28–1.41 | < **0.001** |
| > 3:30 PM | 1.64 | 1.47–1.84 | < **0.001** | 0.73 | 0.70–0.77 | < **0.001** |
| Gender (Male) | 1.16 | 1.01–1.33 | **0.035** | 0.53 | 0.50–0.56 | < **0.001** |
| Day of the week | | | | | | |
| Sunday | Reference | | | Reference | | |
| Monday | 1.03 | 0.89–1.18 | 0.719 | 1.17 | 1.10–1.24 | < **0.001** |
| Tuesday | 0.88 | 0.77–1.02 | 0.086 | 1.30 | 1.22–1.38 | < **0.001** |
| Wednesday | 0.96 | 0.84–1.11 | 0.608 | 1.29 | 1.22–1.37 | < **0.001** |
| Thursday | 0.96 | 0.84–1.10 | 0.597 | 0.97 | 0.91–1.03 | 0.262 |
| Friday | 0.95 | 0.83–1.08 | 0.435 | 1.07 | 1.01–1.14 | **0.019** |
| Saturday | 0.97 | 0.85–1.11 | 0.660 | 1.18 | 1.11–1.25 | < **0.001** |
| Month | | | | | | |
| October | Reference | | | Reference | | |
| November | 0.93 | 0.82–1.06 | 0.265 | 1.45 | 1.36–1.55 | < **0.001** |
| December | 1.01 | 0.86–1.18 | 0.930 | 1.44 | 1.33–1.55 | < **0.001** |
| January | 1.05 | 0.88–1.27 | 0.568 | 1.82 | 1.67–1.98 | < **0.001** |
| Administrative Division | | | | | | |
| Rangpur | Reference | | | Reference | | |
| Dhaka | 0.64 | 0.55–0.75 | < **0.001** | 1.23 | 1.14–1.33 | < **0.001** |
| Khulna | 0.77 | 0.68–0.87 | < **0.001** | 1.08 | 1.02–1.14 | **0.005** |
| Rajshahi | 1.00 | 0.82–1.23 | 0.980 | 1.55 | 1.43–1.69 | < **0.001** |
| Venue Type | | | | | | |
| Educational institutions | Reference | | | Reference | | |
| Other | 1.06 | 0.78–1.48 | 0.780 | 1.83 | 1.60–2.09 | < **0.001** |
| Farmer house | 0.91 | 0.75–1.09 | 0.297 | 4.88 | 4.52–5.28 | < **0.001** |
| Market place | 1.15 | 0.96–1.37 | 0.119 | 0.31 | 0.29–0.34 | < **0.001** |
| Religious institutions | 0.65 | 0.49–0.87 | **0.003** | 4.55 | 4.05–5.11 | < **0.001** |
| Roadside venues | 0.90 | 0.72–1.12 | 0.350 | 1.00 | 0.89–1.11 | 0.9380 |
| Shop | 0.74 | 0.57–0.95 | **0.018** | 2.72 | 2.44–3.03 | < **0.001** |
| Tea-stall | 0.93 | 0.76–1.15 | 0.528 | 2.44 | 2.23–2.67 | < **0.001** |
| Union parishad campus | 0.99 | 0.72–1.37 | 0.942 | 0.47 | 0.39–0.56 | < **0.001** |

[a] Adjusted by years of education of the trainer

[b] Dispersion parameter estimate: 2.798 (SE: 0.118).

spaces. Interactions with men beyond those who are immediate family members is often also frowned upon [36–40]. Under such challenging circumstances, it is important to avoid practices that may actually impede women's participation. Our results demonstrate how factors that increase access and participation in agricultural extension in general can also inhibit women's access and participation (e.g., male-led training events). Fortunately, not all factors affect both overall training attendance and the odds of female participation (e.g., the day of the week affects the odds of females attending but not general attendance). Combined with an understanding of the cultural sensitivities described above, our results suggests that extension

services can nonetheless be designed to improve female attendance without negatively impacting general turnout.

Findings in the present study are consistent with the literature. For example, the finding that only approximately 19% of training attendees were females is consistent with other research, finding disproportionately less participation in extension by females, ranging from 6.9% to 25.3.% in Bangladesh [22, 41]. Also, the present study confirms that gender inequality exists within extension services. The percentage of female trainers in this study (slightly less than 10%) concurs with other studies, ranging from 7% in Bangladesh, to 23.3% in Rwanda, to 33% in Kenya [42–44]. Our results also support other findings that women participants tended to be more comfortable in settings led by other women than by men [44–46], and that female-led training increases women's participation [45–50]. The increased presence of women agents in extension systems is something that has long been advocated and our results show that the presence of women agents clearly makes a statistical difference in women's participation. However, since in our study attendees may not have known the gender of the trainer until arriving at the event, further research is needed in order to understand whether this was a factor in the decision to stay or leave.

Situating these results within the on-going and evolving science of agricultural extension is of utmost importance. As Cook, Satizábal, and Curnow [47] point out in their comprehensive review of extension literature from the 1950s to the present, the recent shift toward systems thinking in the analysis of agricultural effectiveness has shed new light on the heretofore understudied role of socio-political factors in limiting access to agricultural knowledge, especially for women. The tendency of extension science to rely on what Cook et al. [47] describe as the "rendering technical" of extension services, i.e., the reduction of innovation-to-adoption processes to the effectiveness of the technologies introduced or the knowledge communication systems that accompany them, without taking into account the exogenous, mostly socio-political system factors that constrain these processes, is what has led to the less than effective results of agricultural extension systems.

This weakness has been highlighted by recent studies to understand females' participation in agricultural training programs in Bangladesh. For example, Mamun-ur-Rashid et al. [22] observed in their study of females' participation in extension services in Bangladesh, women did not "attend locally arranged extension programs due to the mismatch with their free time schedule. Extension programs [were] generally arranged at times when the women remain busy with household chores" (p. 102). At 9.7%, the number of female trainers was far lower than the proportional participation of females in the agricultural workforce. Females are thought to participate more and retain more when they are able to interact with a female extension agent [45, 48, 49]. Distance from home may also be an important factor impacting female turnout [22], a parameter we were unable to address in the current study.

The issues of women's access to extension services and the agricultural knowledge therein conveyed in given places and at given times may also play a future role in gender balance in such training programs. For example, Rubin, Ferdousi, Parvin et al. [50] identifies five general barriers to female participation in agricultural activities: (1) differences in physical strength, (2) lack of access to resources, (3) lack of skills, knowledge, or experience—this is especially confirmed by Mamun-ur-Rashid et al. [22]—as well as (4) lack of social mobility due to household responsibilities, and (5) fear of dishonor or disrespect. Rubin et al. [50], who studied extension services in Bangladesh specifically do allude to "social norms around time and place" (p. 16) that constrain women's mobility and twice includes quotations from participants who refer to women's access to services: one male participant noted that the extensive and wide-ranging driving he has to do for his farm work would be "not at all convenient for

women" [50, p.17]. Another added that women might not "attend locally arranged extension programs due to the mismatch with their free time schedule" (p. 102).

In this regard, two of our results remain unexplained. Our data suggests that both Tuesday and Wednesday tend to increase female participation more than the other days and that there is clear evidence to suggest that both Sunday and Thursday events decrease female participation. The reasons for these female preferences are unclear but could be related to Sunday being the first day in the work week and Thursday being the last, and that these two days are less convenient for females. However, further research is needed to understand the impact that day of the week has on the likelihood of females participating in training. There is a substantial body of research on women's time allocation in rural Bangladesh [51, 52]; however, these studies address only the allocation of time in the aggregate but do not discuss the question of day of the week time allocation. This is an area that would merit additional research.

In addition, there was a clear trend toward greater participation of women in the later months of the program than for the earlier months, with January being the month of highest participation. We are not certain whether this result was due to learning on the part of the program implementers to improve their outreach to women, or whether seasonal factors may have played a role. In the agricultural cycle in Bangladesh, January comes at the end of the Aman growing cycle which is the time of greatest crop production in Bangladesh, with rice being the principal crop [53, 54]. According to colleagues, this is a time of year when many of those involved have more leisure, depending, of course, on when they choose to plant their winter crops. This may have been a factor in the increased participation but would require further research to confirm.

Women's not-unjustified desires to avoid unpleasant threats of harassment, dishonor, or disrespect [50] can be understood as part of the socio-political landscape of which they are a part. As one interviewee noted, "The reason [that women do not come to buy fertilizer at the shop] is men are forward in every place (work)" [50, p. 17]. However, two interviewees noted that this desire to avoid unpleasantness could be overcome if women did these activities in groups or if there were other women around [50, p. 17]. As such, this group element likely makes it more convenient for women to participate in extension training and may also illuminate the reason for fewer women participating in male-led trainings (as compared to female led events) and larger-sized mixed events in certain venues. This is consistent with the statistical findings of Kondylis et al. [46] in Mozambique showing that women's participation in extension services is increased when extension events are led by females. As they conclude, "This result suggests female messengers may increase female farmer awareness of the technology and hence their demand for information" (p, 446). As a recent study by Kumar et al. [55] demonstrated, women in five Indian states were more empowered and more engaged in government offered extension services when they worked in groups with other women. Making extension events accessible in terms of time of day and venue would increase women's participation.

Another conclusion of this study is the potential for ICT to help bridge the access gap. In the events described in this study, ICT was deployed through in-person extension training events and ICT is increasingly being delivered through online channels, which can potentially bridge gender gaps for agricultural extension services by lowering social barriers [23]. The portability of digitized extension training materials decreases the need for trained extension agents to be involved in the dissemination of key agricultural information. In its offline format, it can be taken just about anywhere, reaching groups that have until now been difficult to reach. For example, ICT educational animations delivered online could avoid any mixed-gender interaction prohibitions while maintaining high curricular standards, learning transfer, and information up-take [56, 57]. At the same time, offline use of the same resources can be taken from farmer's house to farmer's house, venues that are friendly to female agricultural workers. They

can also be viewed in private, reducing even further the barriers that might affect participation. Despite extensive literature on females' access to extension services [58–62], less is known about rural females' experiences around ICT online access generally [63], including for agricultural extension services.

Perhaps the most important result of this study, however, is the potential that big data and machine learning represent for transforming the way that we understand and practice extension [64–67]. The UN has referred to our current age as the "Industrial Revolution of Data" [68]. It represents the opportunity of turning imperfect, complex, often unstructured data into actionable information. In this case, for example, more than 130,000 training participants and 1,070 data points has allowed for precise, statistically verifiable insights into the factors affecting women's participation in agricultural extension in Bangladesh. As extension services are rendered and data is collected on participants, machine learning would allow the modeling of factors affecting participation of women (or any group for that matter) to become more and more sophisticated and precise. Combining individual demographic information with more and more precise geographical locations may, on a routine basis, help us to define more precise model-driven extension strategies.

Finally, the current paper explores factors that drive general population and female participation in ICT-enhanced agricultural training. As the results demonstrate, optimizing situational factors is not trivial given that some factors affect both general turnout and odds of females attending (often in different directions), while some factors only affect the odds of female attending but not the general population. Given the complex interaction between general and female attendance across several factors that vary in their influence, sophisticated modeling is needed to optimize this multivariable problem. Moreover, factors may be time-varying, (e.g., female attendance is improving over time); therefore, adaptive modeling will be needed, which will require continuous sampling of data to update models. Extension services should consider building in analysis tools that allow ICT-enhanced training to learn from past interventions to be adaptive for subsequent interventions.

## Study limitations

There were several limitations with the study that should be considered. First, this was a convenience-sampled, observational study. Second, only limited information associated with session attendance instead of individual participant details were available and therefore underlying factors such as age, education level, religious affiliation, and socio-economic status, among others, of the persons attending the session could not be assessed. Furthermore, our models assume that the attending (or not) of an individual person is conditionally independent of other unobserved factors. Additionally, not all Bangladesh Divisions were represented in the study. This was a result of the nature of the training sessions administered: rather than focus on covering the whole country, they were concentrated into project areas in which maize, rice, and wheat are intensively grown. However, given that half of the divisions in Bangladesh were included, and that there was a large geographical coverage within each Division, the data collected still provide opportunity to explore potential differences for female training participation. Finally, it is important to note that the results should be viewed as specific to this particular type of training approach. In the future, other forms of information dissemination (e.g., social media and television campaigns) could also be studied to determine if gender specific preferences exist.

## Conclusions

In conclusion, the study found gender disparities in terms of female participation in these training sessions. Therefore, even though ICT-enhanced educational programs have the

potential for scaling to large numbers of people (e.g., in this study to over 130,000 people), they could also generate gender disparities. ICT, as a technology can be gender neutral; however, how it is deployed may either amplify or minimize gender biases. The recent agricultural extension policy published by the Bangladesh Ministry of Agriculture [69] highlights both gender inclusion and e-extension efforts as part of their priorities. The inclusion of women is mentioned seven times in the document (§§ 1.3, 1.4, 3.2, 4.1, 4.3, and 5). Especially the last of these references speaks to the Ministry's commitment to "taking appropriate expansion measures to increase the participation of women at all levels of modern agricultural production and marketing system" (p. 14). Likewise, support for e-agriculture is an important part of the Ministry's strategy (§3.2). Based on our findings we make the following recommendations.

## Make data collection on agreed-upon indicators a standard part of extension training

The potential for refining extension services to meet the goals of gender inclusiveness is great if data collection on a slate of key indicators is made a routine part of extension training. The list of indicators will undoubtedly change over time as more awareness of factors influencing participation decisions become known. This would allow for a process of continual analysis and adjustment based on statistically derived information on important factors for participation in general, and female participation in particular. Unfortunately, however, few national extension programs conduct research to adaptively manage their practices to reach greater numbers of farmers, nor is gender or social inclusion incorporated as a core priority [57, 70]. Significant changes in the ways in which extension programs are designed—so that they both provide, but also collect, analyze, and interpret data, and adjust their practices accordingly— are needed in response to these challenges, and in particularly with respect to improving social inclusion [71]. The Bangladesh *National Agricultural Extension Policy* [69] already commits itself to automatic data collection on climate as part of its strategy (§3.2). A similar commitment should be made to collecting information on women's participation for the purpose of improving women's access to agricultural information. This should be required not only of state-sponsored extension, but also of NGO's operating in country whose activities include extension. Participation in the collection of key data would allow for governments to make key policy decisions in order to expand access to and the effectiveness of extension training.

## Use big data to improve the effectiveness of extension services, and specifically gender inclusiveness

Our analysis also shows the value of big data for identifying circumstances where female participation can be optimized for future ICT-enhanced educational programs in the regions of Bangladesh where these initial interventions occurred. However, this study also demonstrates the need for analysis of scaled ICT-programs to determine situational parameters that will optimize female participation. Future work needs to be performed to understand parameters across communities, cultures, and countries towards optimizing female participation in scaled ICT-enhanced training. Such big data collection and analysis approaches should be "hardwired" (i.e., as part of the design process) into such scaling efforts, as gender bias may be time-varying, thus requiring continual adaption of scaled ICT-enhanced training. Hardwiring of data collection and subsequent analysis of scaling efforts, therefore, is critical for the development community to pragmatically offer evidence-based equal access and opportunity to women.

### Make female participation in extension training a priority

Finally, in order to enhance development outcomes, maximizing female participation in extension trainings should be a priority. A recent study [5] showed that females participate in most components of rice production in Bangladesh, although their activities tended to be mostly concentrated in the area of post-harvest (cleaning, drying, storing, bagging, income management, and sorting). Although female participation undoubtedly varies from crop to crop, it is safe to assume that it is significant across the commodity groups. This alone should warrant specific strategies to provide extension services that are developed specifically for female participants in the production value chain of different commodities, either jointly with males, or separately to address those parts of the value chain that are supported mostly by females. This should include specific steps to increase the number of women extension agents in order to increase the odds of women's participation in extension training events like those described in this study. Only when extension services are truly inclusive will it be possible to maximize agricultural potential.

## Supporting information

**S1 File. Minimal data set.** This is the minimal data set requested by the editor. This is not to be included in the manuscript.
(XLSX)

## Author Contributions

**Conceptualization:** John William Medendorp, N. Peter Reeves, Timothy J. Krupnik, Anne N. Lutomia, Barry Pittendrigh, Julia Bello-Bravo.

**Data curation:** John William Medendorp, N. Peter Reeves, Victor Giancarlo Sal y Rosas Celi, Md. Harun-ar-Rashid, Timothy J. Krupnik.

**Formal analysis:** N. Peter Reeves, Victor Giancarlo Sal y Rosas Celi.

**Funding acquisition:** John William Medendorp, Timothy J. Krupnik, Barry Pittendrigh, Julia Bello-Bravo.

**Investigation:** Md. Harun-ar-Rashid, Anne N. Lutomia.

**Methodology:** N. Peter Reeves, Victor Giancarlo Sal y Rosas Celi, Barry Pittendrigh, Julia Bello-Bravo.

**Project administration:** Md. Harun-ar-Rashid, Barry Pittendrigh, Julia Bello-Bravo.

**Resources:** Julia Bello-Bravo.

**Visualization:** N. Peter Reeves, Victor Giancarlo Sal y Rosas Celi, Barry Pittendrigh.

**Writing – original draft:** John William Medendorp, N. Peter Reeves, Victor Giancarlo Sal y Rosas Celi, Anne N. Lutomia, Barry Pittendrigh, Julia Bello-Bravo.

**Writing – review & editing:** John William Medendorp, N. Peter Reeves, Victor Giancarlo Sal y Rosas Celi, Md. Harun-ar-Rashid, Timothy J. Krupnik, Anne N. Lutomia, Barry Pittendrigh, Julia Bello-Bravo.

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
