## [Decision Letter · Decision Letter 0]

11 Jan 2022

PONE-D-21-29046Large-scale rollout of information and communication technology-enhanced extension training in Bangladesh demonstrates challenges and opportunities towards inclusive gender participationPLOS ONE

Dear Dr. Bello-Bravo,

Thank you for submitting your manuscript to PLOS ONE. After careful consideration, we feel that it has merit but does not fully meet PLOS ONE’s publication criteria as it currently stands. Therefore, we invite you to submit a revised version of the manuscript that addresses the points raised during the review process.

We look forward to receiving your revised manuscript.

Kind regards,

Bidhubhusan Mahapatra, Ph.D.

Academic Editor

PLOS ONE

Journal Requirements:

“This publication was made possible in part by internal funds from Michigan State University and Purdue University. The fall armyworm animation and scaled extension activities were supported by funds provided to CIMMYT by the Borlaug Higher Education for Agricultural Research and Development Program (BHEARD; USAID award # AID-BFS-G-11-00002) at Michigan State University and the Bill and Melinda Gates Foundation support for the Cereal Systems Initiative for South Asia (CSISA; USAID award #BFS-IO-17-00005 and BMGF isINV-009787). Its contents are solely the responsibility of the authors and do not necessarily represent the official views of USAID, BHEARD, CSISA, or the Bill and Melinda Gates Foundation. This study was deemed exempt by Michigan State University Biomedical and Health Institutional Review Board (IRB 00004626).”

“This publication was made possible in part by internal funds from Michigan State University and Purdue University [JBB & BRP]. The fall armyworm animation and scaled extension activities were supported by funds provided to CIMMYT by the Borlaug Higher Education for Agricultural Research and Development Program (BHEARD; USAID award # AID-BFS-G-11-00002) at Michigan State University [JWM] and the Bill and Melinda Gates Foundation support for the Cereal Systems Initiative for South Asia (CSISA; USAID award #BFS-IO-17-00005 and BMGF isINV-009787)[TJK].

USAID: https://www.usaid.gov

BMGF: https://www.gatesfoundation.org

Its contents are solely the responsibility of the authors and do not necessarily represent the official views of USAID, BHEARD, CSISA, or the Bill and Melinda Gates Foundation.

No sponsors or funders played any role in the study design, data collection and analysis, decision to publish, or preparation of the manuscript.”

Additional Editor Comments (if provided):

An useful paper with some interesting findings. I agree with the observations made by two reviewers and suggest authors revise the paper accordingly. Some of my observations are as follows:

1. Please follow the PLOS one guideline to arrange the different sections of the paper. For example, the limitation should be part of the discussion NOT as part of results.

2. Some of the details provided as part of descriptive analysis should actually be part of method section. Similarly, the data section instead of telling about data, it details the program. I suggest renaming the section as intervention and include a new section data where describe how the data was gathered and processed? How was the quality control done?

3. Justify the use of statistical methods, specifically negative binomial.

4. For timing of the authors switched from 4 category to 3 category and justified it through statistical test. I am wondering if authors did for other categorical variables. If yes, please include the test results as supplementary files and write about in the analysis section.

5. The English language needs a thorough review.

Reviewers' comments:

Reviewer's Responses to Questions

**Comments to the Author**

1. Is the manuscript technically sound, and do the data support the conclusions?

Reviewer #1: Yes

Reviewer #2: Yes

2. Has the statistical analysis been performed appropriately and rigorously? 

Reviewer #1: Yes

Reviewer #2: Yes

3. Have the authors made all data underlying the findings in their manuscript fully available?

Reviewer #1: Yes

Reviewer #2: Yes

4. Is the manuscript presented in an intelligible fashion and written in standard English?

Reviewer #1: Yes

Reviewer #2: Yes

5. Review Comments to the Author

Reviewer #1: This manuscript covers an important area of ICT-based extension training in Bangladesh towards gender participation and gender inequalities in women’s access and participation, which has become increasingly important for women's participation in agricultural activities in Bangladesh. Furthermore, the manuscript also explores and compares the gendered impacts from the independent variables and how to increase women’s involvement in ICT-based extension training.

I found this study interesting.

The manuscript is worthy of publication in this journal, but minor amendments listed below need to be addressed.

Nonetheless, my concrete comments are given below by the manuscript section-wise, which are needed to improve the manuscript quality.

Overall comments:

a. The manuscript presentation is well enough; however, the sentence structure needs to write in a simple form (especially in the methods section) to be considered a full-length paper and for better readership.

b. Authors should read the whole manuscript several times after correcting the comments below section-wise with the total concentration. And then, the whole manuscript needs to refine/rewrite/reorganize sentences to maintain the sentence consistency and subsequently easy understanding for scientific readership from the preceding sentence to the running sentence.

c. References should be checked for improving with the currently published article.

d. It is not fair not to insert the line numbers in the manuscript; however, authors should insert line numbers in the revised manuscript version.

Abstract

1. It is well written, and abstract content has been found systematically organized. However, the sentence before the last sentence needs to improve.

Introduction

2. The introduction should be written systematically; it should be organized like current trends of gender participation in ICT-based extension training, gender inequalities in extension training, parameters limitation for women’s participation, variables impacts on women’s participation, review on how to improve women’s assess in participation, research gap, hypothesis, and objectives, etc.

Methods

3. On page 5, ‘this was’ should be deleted from the first sentence.

4. There is a blue color sign in Figure 1B? it should be replaced with the new one.

5. Authors should include some valuable pictures and a flow sheet of data collection in the data collection subsection. It gives a clear illumination about the methods section.

6. Authors should cover some theory behind this study and equations to analyze the data.

Results and Discussion

Please recheck the results and calculation sheets to delineate the exact and accurate amount to be sure again.

7. For descriptive data paragraph in the results chapter, the structure and meaning of the sentence are proper, but the way of presenting the sentence is not correct need to improve sentence structure more clearly as a simple form for better readership.

8. The discussion is well written.

Limitations

9. This section has been written well; however, the authors should mention another limitation- ICT enhanced extension training for empowering women’s skills to be involved more in agricultural works by using the television media.

Conclusion

10. The conclusion is well written in a descriptive way, not written in numerative and contains recommendations.

Reviewer #2: The study provides critical insights into where and when Bangladeshi agricultural communities are more likely to access and participate in extension training services. Moreover, it also demonstrates that ICT-enhanced educational programs have both the potential for scaling to large numbers of people and the potential to amplify gender disparities in terms of female participation. The study satisfied almost all the criteria to publish as an original research paper.

However, handing the followings would improve its quality:

1. I would like to suggest the authors to simplify the title.

2. In the abstract, please write eight administrative divisions, not division.

3. At the end of page#4, For instance, educational ICT…….attended events requires reference. Also, for the other lines of the paragraph, at least two more references are required.

4. Methods: This was mentioned two times (please check page#5)

5. In page#6, Majumder, 2013 is very old information for rice crop. Please, check latest studies done by BRRI researcher for updated information on rice.

6. In the data collection procedure, please mention the number enumerators per session and the time spent per respondent while interview.

7. Data analysis: Why binomial regression with logit? Why not other models? such as OLS, MLE, Frictional logit, probit, etc. Please, justify the utilization of statistical tools.

8. Descriptive data: 10 sessions were missing information should be 10 sessions had missing information.

9. In page#9 paragraph#2, your finding described that if the trainer is male, the total participation increased by 16%, but female participants reduced by 47%. This is interesting, but did the participants new about the gender of the trainer before the training session? In general, the participants do not know before the session start about the trainer and do not impact on the number of participants.

10. You also find out that women are more likely to attend in the training on Tuesday. But, why Tuesday? Why less likely to other days? Explaining the reasons in the discussion would generate important policy guidelines.

11. Also, why women likely to attend in January than October? Please, explain the reasons in the discussion.

12. What I was expecting at the end of this paper, the policy options for enhancing agricultural extension (most importantly for the development of women friendly agricultural extension policy) is lacking in this study. I would like to request the authors to review the agricultural extension policy of Bangladesh and suggest to fill up the gaps in the light of findings of this study.

6. PLOS authors have the option to publish the peer review history of their article (what does this mean?). If published, this will include your full peer review and any attached files.

Reviewer #1: **Yes: **Md Mashiur Rahman

Reviewer #2: **Yes: **Mohammad Chhiddikur Rahman

---

## [Author Response · Author response to Decision Letter 0]

1 Apr 2022

We have included a letter with a point by point response to the editor and the reviewers.

---

## [Decision Letter · Decision Letter 1]

2 May 2022

PONE-D-21-29046R1Large-scale rollout of extension training in Bangladesh: Challenges and opportunities for gender-inclusive participationPLOS ONE

Dear Dr. Bello-Bravo,

Thank you for submitting your manuscript to PLOS ONE. After careful consideration, we feel that it has merit but does not fully meet PLOS ONE’s publication criteria as it currently stands. Therefore, we invite you to submit a revised version of the manuscript that addresses the points raised during the review process. Please look at the some of the comments which one of the reviewers have asked for more clarity.

We look forward to receiving your revised manuscript.

Kind regards,

Bidhubhusan Mahapatra, Ph.D.

Academic Editor

PLOS ONE

Journal Requirements:

Reviewers' comments:

Reviewer's Responses to Questions

**Comments to the Author**

1. If the authors have adequately addressed your comments raised in a previous round of review and you feel that this manuscript is now acceptable for publication, you may indicate that here to bypass the “Comments to the Author” section, enter your conflict of interest statement in the “Confidential to Editor” section, and submit your "Accept" recommendation.

Reviewer #1: All comments have been addressed

Reviewer #2: All comments have been addressed

2. Is the manuscript technically sound, and do the data support the conclusions?

Reviewer #1: Yes

Reviewer #2: Yes

3. Has the statistical analysis been performed appropriately and rigorously? 

Reviewer #1: Yes

Reviewer #2: Yes

4. Have the authors made all data underlying the findings in their manuscript fully available?

Reviewer #1: Yes

Reviewer #2: Yes

5. Is the manuscript presented in an intelligible fashion and written in standard English?

Reviewer #1: Yes

Reviewer #2: Yes

6. Review Comments to the Author

Reviewer #1: I am pleased that the authors are responded to all the comments. The response to the comments provided by the authors is sufficient which could be considered to publish this manuscript and the response to the comments for the section of the introduction, materials and methods, results, discussion, conclusion and reference is, however, consistently organized and well written.

Remember that when proofreading will be performed, everything should be corrected.

Reviewer #2: Comments on the revised manuscript

1. I found the authors have addressed all the comments and tried to fix them up. However, I am not fully satisfied with the response to comments# 29, 30, and 31. The authors did not get information from their survey to meet these comments and demanded further study/investigation required for that. Mentioning the limitations of the study and leaving these for further investigation is scientific. However, I would like to request to search in the literature if the author can find any supporting information to support/explain these findings.

2. I also would like to make the policy implication more gorgeous. The study has enough resources to critique the national agricultural extension policy (NAEP) 2020 (in Bengali). Whether the NAEP emphasizes training the women participants or not. How about the ICT-based training? In the social aspect, do the NAEP realizes the importance of female trainer for enhancing female participants in the training program? If the authors find these issues addressed in the NAEP, can mention them in the justification of policy implication section. If these are found gaps in the NAEP, can suggest to include in the implementation guideline of NAEP.

Finally, I found the paper improved significantly and the editor may accept it with minor revisions.

7. PLOS authors have the option to publish the peer review history of their article (what does this mean?). If published, this will include your full peer review and any attached files.

Reviewer #1: **Yes: **Md Mashiur Rahman

Reviewer #2: **Yes: **Mohammad Chhiddikur Rahman

---

## [Author Response · Author response to Decision Letter 1]

8 Jun 2022

The response to reviewers has also been attached to this submission:

Dear editor and reviewers,

We thank you for your thoughtful comments. We have made changes to the paper based on your comments (see the point-by-point response below). Our comments are in italics. Parts taken from the paper are in quotation marks (“ ”), and new sections added are underlined.

Responses to Reviewers Comments

Reviewer #1:

I am pleased that the authors are responded to all the comments. The response to the comments provided by the authors is sufficient which could be considered to publish this manuscript and the response to the comments for the section of the introduction, materials and methods, results, discussion, conclusion and reference is, however, consistently organized and well written.

Remember that when proofreading will be performed, everything should be corrected.

Thank you for your review. We are grateful for your constructive engagement with the editing process.

Reviewer #2: Comments on the revised manuscript

1. I found the authors have addressed all the comments and tried to fix them up. However, I am not fully satisfied with the response to comments# 29, 30, and 31. The authors did not get information from their survey to meet these comments and demanded further study/investigation required for that. Mentioning the limitations of the study and leaving these for further investigation is scientific. However, I would like to request to search in the literature if the author can find any supporting information to support/explain these findings.

Thank you for that request. We realized upon reviewing the uploaded documents, that we had not uploaded the correct version of the paper. We are including our responses to the previous version as well as these new comments so that you can see how the paper has changed. 

29. In page#9 paragraph#2, your finding described that if the trainer is male, the total participation increased by 16%, but female participants reduced by 47%. This is interesting, but did the participants new about the gender of the trainer before the training session? In general, the participants do not know before the session start about the trainer and do not impact on the number of participants. 

We have conducted a thorough review of the relevant literature as requested by the reviewer. In addition, we have engaged extensively with colleagues in Bangladesh in order to discern some of the possible answers to the reviewer’s questions and concerns. We devote a significant portion of the discussion section addressing the literature on women’s preference for female-led events. We have amended the text accordingly: 

"Our results also support other findings that women participants tended to be more comfortable in settings led by other women than by men [45-46, 44], and that female-led training increases women’s participation [45- 50]. The increased presence of women agents in extension systems is something that has long been advocated and our results show that the presence of women agents clearly makes a statistical difference in women’s participation. However, since in our study attendees may not have known the gender of the trainer until arriving at the event, further research is needed in order to understand whether this was a factor in the decision to stay or leave."

30. You also find out that women are more likely to attend in the training on Tuesday. But, why Tuesday? Why less likely to other days? Explaining the reasons in the discussion would generate important policy guidelines.

Unfortunately, the study was not designed to address why preferences in days exists and there is no clear insights from the literature. So any explanations for women training preferences on Tuesday (and Wednesday) would be speculation. However, at the request of the reviewer, we have added some additional discussion on this topic in the text. 

“In this regard, two of our results remain unexplained. Our data suggests that both Tuesday and Wednesday tend to increase female participation more than the other days and that there is clear evidence to suggest that both Sunday and Thursday decrease female participation. The reasons for these female preferences is unclear but could be related to Sunday being the first day in the work week and Thursday being the last and that these two days are less convenient for females. However, further research is needed to understand the impact that day of the week has in the likelihood of females participating in training. There is a substantial body of research on women’s time allocation in rural Bangladesh [51, 52]; however, these studies address only the allocation of time in the aggregate but do not discuss the question of day of the week time allocation. This is an area that would merit additional research.”

We have also added two new references:

“51. Khandker, S. R. (1988). Determinants of women's time allocation in rural Bangladesh. Economic Development and Cultural Change, 37(1), 111-126.

52. Islam, F. B., & Sharma, M. (2022). Socio-economic determinants of women’s livelihood time use in rural Bangladesh. GeoJournal, 1-13.”

31. Also, why women likely to attend in January than October? Please, explain the reasons in the discussion. 

We added text to indicate that we do not have an explanation for this finding at this time: 

“In the agricultural cycle in Bangladesh, January comes at the end of the Aman growing cycle which is the time of greatest crop production in Bangladesh, with rice being the principle crop [53-54]. According to colleagues, this is a time of year when many of those involved have more leisure, depending, of course, when they choose to plant their winter crops.”

We have also added two new references:

“53. Sultana, S., Khan, M. A., Hossain, M. E., Prodhan, M. M. H., & Saha, S. M. (2022). Yield gap, risk attitude, and poverty status of aman rice producers in climate-vulnerable coastal areas of Bangladesh. Journal of Agricultural Science and Technology, 0-0.

54. Al Mamun, M. A., Nihad, S. A. I., Sarkar, M. A. R., Aziz, M. A., Qayum, M. A., Ahmed, R., ... & Kabir, M. S. (2021). Growth and trend analysis of area, production and yield of rice: A scenario of rice security in Bangladesh. PloS one, 16(12), e0261128.”

2. I also would like to make the policy implication more gorgeous. The study has enough resources to critique the national agricultural extension policy (NAEP) 2020 (in Bengali). Whether the NAEP emphasizes training the women participants or not. How about the ICT-based training? In the social aspect, do the NAEP realizes the importance of female trainer for enhancing female participants in the training program? If the authors find these issues addressed in the NAEP, can mention them in the justification of policy implication section. If these are found gaps in the NAEP, can suggest to include in the implementation guideline of NAEP.

We have reviewed the National Agricultural Extension Policy and have incorporated citations and added the following text to support our recommendations:

“The recent agricultural extension policy published by the Bangladesh Ministry of Agriculture [69] highlights both gender issues and e-extension efforts as part of their priorities. The inclusion of women is mentioned seven times in the document (§§ 1.3, 1.4, 3.2, 4.1, 4.3, and 5). Especially the last of these references speaks to the Ministry’s commitment to “taking appropriate expansion measures to increase the participation of women at all levels of modern agricultural production and marketing system” (p. 14). Likewise, support for e-agriculture is an important part of the Ministry’s strategy (§3.2).”

“The Bangladesh National Agricultural Extension Policy [69] already commits itself to automatic data collection on climate as part of its strategy (§3.2). A similar commitment should be made to collecting information on women’s participation for the purpose of improving women’s access to agricultural information.”

“This should include specific steps to increase the number of women extension agents in order to increase the odds of women’s participation in extension training events like those described in this study.”

Finally, I found the paper improved significantly and the editor may accept it with minor revisions.

Thank you for your kind comments.

---

## [Editor Report · Decision Letter 2]

15 Jun 2022

Large-scale rollout of extension training in Bangladesh: Challenges and opportunities for gender-inclusive participation

PONE-D-21-29046R2

Dear Dr. Bello-Bravo,

We’re pleased to inform you that your manuscript has been judged scientifically suitable for publication and will be formally accepted for publication once it meets all outstanding technical requirements.

Kind regards,

Bidhubhusan Mahapatra, Ph.D.

Academic Editor

PLOS ONE
---

## [Editor Report · Acceptance letter]

21 Jun 2022

PONE-D-21-29046R2 

Large-scale rollout of extension training in Bangladesh: Challenges and opportunities for gender-inclusive participation 

Dear Dr. Bello-Bravo:

I'm pleased to inform you that your manuscript has been deemed suitable for publication in PLOS ONE. Congratulations! Your manuscript is now with our production department. 

Kind regards, 

on behalf of

Dr. Bidhubhusan Mahapatra 

Academic Editor

PLOS ONE